# Deterministic and Probabilistic Dietary Exposure Assessment to Deoxynivalenol in Spain and the Catalonia Region

**DOI:** 10.3390/toxins14070506

**Published:** 2022-07-20

**Authors:** Jose A. Gallardo, Sonia Marin, Antonio J. Ramos, German Cano-Sancho, Vicente Sanchis

**Affiliations:** 1Department of Food Technology, AGROTECNIO-CERCA Center, University of Lleida, 25198 Lleida, Spain; joseantonio.gallardo@udl.cat (J.A.G.); sonia.marin@udl.cat (S.M.); antonio.ramos@udl.cat (A.J.R.); vicente.sanchis@udl.cat (V.S.); 2Laboratoire d’Etude des Résidus et Contaminants dans les Aliments, Oniris (LABERCA), UMR1329, Institut National de Recherche pour l’Agriculture, l’Alimentation et l’Environnement (INRAE), 101 Route de Gachet, 44300 Nantes, France

**Keywords:** deoxynivalenol, exposure assessment, cereal-based food, *Fusarium*

## Abstract

Deoxynivalenol (DON) remains one of the most concerning mycotoxins produced by the *Fusarium* genus due to the wide occurrence in highly consumed cereal-based food and its associated toxicological effects. Previous studies conducted in Spain and other European countries suggested that some vulnerable groups such as children could be exceeding the tolerable daily intakes. Thus, the aim of this study was to conduct a comprehensive and updated dietary exposure assessment study in Spain, with a specific analysis in the region of Catalonia. Cereal-based food samples collected during 2019 were analysed using liquid chromatography coupled to tandem mass spectrometry for multi-mycotoxin detection including DON and its main metabolites and derivatives. Consumption data were gathered from the nation-wide food surveys ENALIA and ENALIA2 conducted in Spain, and a specific survey conducted in Catalonia. The data were combined using deterministic and semi-parametric probabilistic methods. The results showed that DON was widely present in cereal-based food highly consumed in Spain and the Catalonia region. Exposure to DON among the adult population was globally low; however, among infants aged 3–9 years, it resulted in the median of 192 ng/kg body weight/day and the 95th percentiles of 604 ng/kg body weight/day, that would exceed the most conservative safety threshold for infants. Bread and pasta were the main contributing foodstuffs to the global exposure to DON, even among infants; thus, those foods should be considered a priority for food control or to develop strategies to reduce the exposure. In any case, further toxicological and epidemiological studies are required in order to refine the safety thresholds accounting for the sensitivity of the infant population.

## 1. Introduction

Trichothecenes are a group of secondary metabolites with toxicological properties produced in cereal grains by several fungal genus like *Fusarium* [1], and thus some of them can be found in cereal-based food [2]. That is the case of deoxynivalenol (DON) and its derivatives 3-acetyldeoxynivalenol (3-Ac-DON) and 15-acetyldeoxynivalenol (15-Ac-DON) [3]. The main toxic effects of DON observed in animals are due to the inhibition of protein synthesis, resulting in a list of health symptoms such as growth retardation, anorexia, vomiting, immune dysregulation and reproductive effects [4,5,6,7]. Hence, the European Food Safety Authority (EFSA) considered DON, and its derivatives, to be a concerning risk to animal and human health [3]. In turn, the European Commission has enforced regulatory maximum limits on feed and cereal-based food [8].

In order to conduct human risk assessment to DON, the EFSA CONTAM Panel considered reduced weight gain in experimental data as the critical chronic endpoint. The benchmark dose for a 5% response (BMDL05) of 0.11 mg DON/kg body weight (bw)/day was estimated in mice and translated into a tolerable daily intake (TDI) of 1000 ng/kg bw/day after using the default uncertainty factor of 100, for the sum of DON, 3-Ac-DON, 15-Ac-DON and DON-3-glucoside (3GDON) [3]. Nevertheless, due to the associated uncertainties and the lack of epidemiological data, the panel of experts on food contaminants from The French Agency for Food, Environmental and Occupational Health & Safety (ANSES) has considered an additional uncertainty factor of three and established the health-based guidance values (HBGV) of 300 ng/kg bw/day for children [9].

At present, DON can be frequently detected in cereal-based products consumed in European countries such as France, the Netherlands or Spain [2,9,10]. For instance, the occurrence of DON in bread and breakfast cereals from Spain was reported up to 100% and 75%, respectively [2]. The high occurrence of DON in cereal-based food can easily result in high exposure due to the large consumption of these products among the general population, as shown by Narváez et al. (2022). In France, the Total Diet Study showed dietary exposure estimates for infants aged 7–12 months ranging between 98 and 648 ng/kg bw/day for the lower and upper-bound approach, respectively, representing a proportion of children above the safety level of 300 ng/kg bw/day between 8% and 76% [9]. Similar findings were also found in our first study conducted in Catalonia, a north-eastern region of Spain, where we found that the safety levels of DON could be exceeded by at least 10% of the infant population, thus demanding refined probabilistic exposure assessment modelling methods [11].

Unlike biomonitoring-based methods that directly estimate the exposure to mycotoxins at an individual level, dietary modelling computes the probable exposure through dietary intake by combining consumption data and contamination data [12]. This process is often conducted in a tiered manner, using direct/deterministic methods in the preliminary assessment and more refined probabilistic methods when exposure levels may become concerning [13,14].

A recent meta-analytic study conducted in Spain showed that most contamination data available to the evaluation of DON were published between 2010 and 2014 [2], supporting the need for more updated data and refined probabilistic methods. Considering the widespread presence of DON in cereal-based food from Spain and the high exposure levels among some population groups, such as children [2], a follow-up study using probabilistic methods with a special focus on the region of Catalonia was conceived, extending our previous assessments. Thus, the present study aimed (1) to evaluate the occurrence of DON in cereal-based foods; and (2) to estimate the exposure among the population from Spain, and more specifically, Catalonia.

## 2. Results and Discussion

### 2.1. Occurrence of Mycotoxins

Among all the type-B trichothecenes analysed (DON, 3-Ac-DON, 15-Ac-DON and 3GDON), only DON was detected in quantifiable amounts in the analysed samples (Table 1), being that all metabolites were found below the LOD of 2.5 µg/kg. The frequency of detection was 100% for most items, with the exception of cookies and baby food. The highest concentration level was found in loaf bread (314.50 µg/kg) and the highest average value in whole bread (183.52 µg/kg), which showed more than 2-fold the mean levels in white bread (76.71 µg/kg). All samples showed DON concentrations below the levels established by the Commission Regulation (EC), as shown in Table 1 [8]. The present values found in bread were comparable to those previously reported in the same region. For instance, in 2009 the mean values in bread were 247 µg/kg and 68 µg/kg in sliced bread, whereas wholemeal bread was not specifically analysed [11]. These values were also consistent with the occurrence levels of bread from France of 132.1 µg/kg [15], and lower than those found in Lebanon with a mean of 524.17 µg/kg [16].

The levels in cookies and other pastry products (cakes and muffins) were substantially lower than in bread (2–3-fold), in the same line as reported in France where values of 61.9 µg/kg in cookies and 54–55 µg/kg in cake were reported [15]. For other transformed products such as breakfast cereals and pasta, the mean levels (range 66.74–65.55 µg/kg) were close to those found in white bread, and far from the 750 µg/kg established by the EC legislation (Table 1). These values are 2 and 3-fold lower than those concentrations found in the same region in 2009: 190 µg/kg in wheat flakes and 109 µg/kg in corn flakes or 226 µg/kg in pasta [11]. Previous studies in Spain found a mean value of 5 µg/kg of DON in pasta [17], whereas mean values ranging between 29 and 116 µg/kg were reported in pasta from France [15], in biscuits and breakfast cereals from Portugal [18], in biscuits and cookies from Netherlands [10] or pasta from Lebanon [16].

In the case of baby food, DON was detected in only 5 individual samples out of 27, representing different brands, and showing a maximum of 180.8 µg/kg, but most of the contaminated samples were in the range between 5.1 and 16.8 µg/kg and the composite 4.8 µg/kg. All samples were compliant with the current maximum level established by the European legislation set at 200 µg/kg [8]. In our previous study, DON was more frequently detected and at higher concentrations. For instance, 12 out of 30 composite samples pooling 90 individual samples were above LOD and the mean concentration was 131 µg/kg [11]. Our present findings are consistent also with more recent studies conducted in Spain, where mean levels of DON were 37 µg/kg [19], but also in Portugal (37.5–71 µg/kg) or Turkey (98.3 µg/kg); all of them below the maximum limit [1,20].

### 2.2. Consumption of Cereal-Based Foods

The normalized consumption of cereal-based products among adult consumers from the Catalonian region and Spain is shown in Table 2. By far, the cereal-based foodstuffs with the highest percentage of consumers and levels of consumption were bread and pasta. Bread rusks and crackers appeared to be occasional. Whereas the consumption patterns between age groups were quite consistent, some variations were appreciated among regions and/or surveys. For consumers under 18 years, bread was also the product with the highest percentage of consumers, but cookies had a higher percentage for children under 3 years old than for those over 3 years old (Appendix A). The consumption of the main cereal-based foods such as bread and pasta was quite consistent with our previous study conducted in 2009 [11].

### 2.3. Exposure Assessment of Adult Population to Deoxynivalenol

The exposure estimates to DON among the adult population from Spain and the Catalonia region are summarised in Table 3. The mean and median exposure levels were found to be in a range between 117 and 183 ng/kg bw/day, across all age groups and exposure assessment methods. These values are far below the TDI of 1000 ng/kg bw/day for DON, representing a range between 11.7% and 18.3% of the threshold. Small differences were found between age groups, with elders (>60 years/old) being the age group that showed slightly lower estimates. Gender comparisons did not reveal differences (data not shown). The results were quite consistent between estimation methods, with similar estimates from the deterministic and probabilistic methods; nonetheless, some differences were found for the highest percentiles. The exposure estimated in Spain was slightly lower than those exposure estimates in Catalonia. These summary estimates were below the established TDI for DON (1000 ng/kg bw/day), being only the P99 probabilistic exposure of the 41–60 age range estimate of Spanish data close to that value. Previous studies conducted in European countries reported values of DON in a similar range, all below the TDI (summarised in Table 4). For instance, our previous deterministic analysis conducted in the same region showed results in the same line, but different estimates from the probabilistic models [11]. The differences can be explained by the modelling approach, integrating a probabilistic component for the contamination distributions, and resulting in more extreme exposure scenarios. The estimates for other European countries were also aligned, with a mean concentration of 37.3% of TDI reported in France [15], or an exposure between 95.8 ng/kg bw/day and 254 ng/kg bw/day was estimated in ages 7 to 69 years old in the Netherlands [21]. Furthermore, Ostry et al. (2020), using a deterministic method in the Czech Republic, showed mean values of 117 to 163 ng/kg bw/day, overlapping the values of the present study [22]. In all cases, those estimates do not raise concerns about potential health effects among the adult population based on the current TDI. Major differences in exposure estimates were found when we compared our findings with some studies conducted outside of the European Union. For instance, Mahdjoubi et al. (2020) reported a mean intake of a 4920 ng/kg bw/day value above TDI (492% of TDI) in Algeria. That is not the case in China, where similar estimates to the present study were found with a probabilistic method showing a mean exposure level of 300 ng/kg bw/day for adults [23].

The relative contribution of each food product to global exposure is shown in Appendix A. The main contributor to the global exposure in Spain estimates was bread with minor differences between age groups (range 50–55%), followed by sliced bread (14–25%) and pasta (12–14%). The estimates for the Catalonian region also showed that whole bread (30–36%) and white bread (27–34%) were the main contributors, followed by pasta (12–16%).

### 2.4. Infant and Adolescent Exposure Assessment

The exposure estimates of infants up to 3 years old are summarised in Table 5. The median exposure estimates were in the range between 36 and 183 ng/kg bw/day. The impact of left-censorship was modelled in the data from the Catalonia region showing that uncertainty may impact the estimates by a magnitude of 2-fold (e.g., composite scenario vs. upper-bound scenario). The estimates between Spain and the Catalonia region showed similar values, yet the age ranges could not be directly compared. The results also showed that the age group with the largest exposure would be infants aged 3–9 years old with median values of 192–193 ng/kg bw/day representing 19% of the TDI or 64–65% of the HBGV proposed by ANSES. As found for adults, the differences between probabilistic and deterministic estimates were only found in the highest percentiles. For instance, the probabilistic method provided estimates for the 95th and 99th percentile environ 23% larger than those from the deterministic method, respectively. The highest percentiles showed exposure values close to the TDI of 1000 ng/kg bw/day or exceeding the HBGV of 300 ng/kg bw/day.

We detected a single sample of baby food with an outlying high level of contamination of 181 µg/kg of DON which was used to estimate the worst-case scenario for acute dietary exposure, which was contrasted against the acute reference dose (ARfD) of 8000 ng/kg bw/day for acute risk characterisation [3]. The estimates showed median levels of p95 and p99 of 2715, 7024 and 9065 ng/kg bw/day, respectively. These results showed that 1% of the infants could slightly exceed that threshold and thus be at risk of acute health effects.

Among infants and adolescents, the main contributor product to global exposure to DON was also white bread (Appendix A), with small differences among age groups (48–53%). Other relevant contributors were cookies and pasta (12–33%).

In our previous study, the mean exposures of infants from Catalonia were in the ranges between 740 and 900 ng/kg bw/day, substantially larger than the present study due to the higher occurrence levels of DON in 2009 [11]. Other previous studies also showed slightly higher exposure values, as summarised in Table 6. For instance, in France, the estimation of babies under one year old was up to 55.3% of the TDI for DON [9], and in Norway between 30% and 200% of the TDI [24]. In turn, Portuguese studies showed estimates more aligned with our present findings for the infants aged 1–3 years with mean exposures representing 5.4–11.3% of the TDI [18], yet adolescents aged 10–17 years would reach 87% of TDI [27].

The contributions of each food group across ages are displayed in Appendix A, showing that white bread and pasta were the main contributors for all groups, whereas cookies were especially relevant (21%) among those infants aged 0–11 months.

## 3. Conclusions

The present study showed that DON is still widely present in cereal-based food highly consumed in Spain and the Catalonia region; nonetheless, in all cases, the samples were compliant with the maximum levels established by the EU legislation. The comprehensive exposure assessment study showed that estimates for adults were far below the TDI of 1000 ng/kg bw/day, with median values in the range between 105 and 156 ng/kg bw/day with small differences due to region, age or estimation method. The occurrence of DON in infant food was low in terms of frequency of detection and concentration values. Nonetheless, the results showed that the introduction of adult food at early ages can result in potentially high exposure to DON, mainly driven by bread, cookies and pasta. For instance, the exposure levels among the age group of infants aged 3–9 years reached 604 ng/kg bw/day for the 95th percentile, which were close to the TDI or even exceeding the HBGV of 300 ng/kg bw/day. Considering the high impact of bread and pasta on the exposure of DON, special attention should be paid to the related food processing chains in order to decrease the final occurrence in the final products. Further toxicological and epidemiological studies must be required in order to refine the safety thresholds, accounting for the sensitivity of the infant population.

## 4. Materials and Methods

### 4.1. Food Sampling and Preparation

The food sampling, conducted between 24 April 2019 and 14 May 2019, included a total of 347 individual samples, accounting for 220 cereal-based food items industrially produced and 127 samples of artisanal bread (Table 7). The samples were collected from the four Catalonian provinces (Barcelona, Girona, Lleida and Tarragona), a north-eastern region of Spain. Baby food samples were collected by brands, and the other samples were collected from three types of distributors: a hypermarket (group 1), a supermarket (group 2) and a group of local distributors (group 3). Cookies were divided into two groups: the group “Cookies” were only basic cookies; and the group “Other cookies” includes chocolate cookies, cream cookies, butter cookies, tea cookies and digestive cookies. Samples were first dried at 40 °C before being homogenised and milled in a thermomix (TM 3300) obtained from Vorwerk (Wuppertal, Germany). Once milled, the individual samples (except bread and baby food) were mixed per core group in analytical composites, resulting in 27 composites (three composites per group). In the case of bread, 4 composites were obtained by combining the individual samples from each region. For baby food, the 32 individual samples were analysed separately.

### 4.2. Chemical Analysis of Mycotoxins

A commercial internal standard was used: 50.3 µg/mL of deepoxi-deoxynivalenol (DOM) purchased from Sigma Aldrich (St. Louis, MO, USA) to spike composites with 50 ppb of the standard. For the extraction, 2 g of each composite or individual sample was placed in a Falcon tube with 18 mL of the extraction solvent: acetonitrile analytical reagent grade 99.99% purchased from Fischer Scientific (Loughborough, UK), bi-distilled water and formic acid analytical reagent grade 98–100% obtained from Fischer Scientific (Leicestershire, UK) in 60:37:3 proportion, respectively. The Falcon tubes were placed in an orbital shaker minitron obtained from INFORS HT (Bottmingen, Switzerland) at 200 rpm for 30 min. Subsequently, a mixture previously prepared of 4 g of magnesium sulphate anhydrous technical grade 96% obtained from PanReac Applichem (Darmstadt, Germany) with 1 g of sodium chloride reagent grade ≥ 99.5% purchased from Fischer Scientific (Loughborough, UK) was added and manually shaken for one minute and placed again in the orbital shaker for 30 min. Then, the mixture was centrifuged at 3000 rpm for 10 min in a centrifuge purchased from Hettich (Tuttlingen, Germany), and 7 mL of the organic fraction was taken and placed in a glass tube. The solvent was evaporated at 40 °C under a gentle stream of nitrogen. For greasy samples, the mixture was centrifuged at 5000 rpm for 5 min, and then, 10 mL of n-hexane 96% obtained from Scharlab (Sentmenat, Spain) was added and shaken in the orbital shaker again at 200 rpm for 30 min. After that, the mixture was centrifuged again at 5000 rpm for 5 min, and hexane was extracted. The rest of the procedure was the same as for the other samples. The extraction method was performed as requested by the laboratory of the Centre of Excellence in Mycotoxicology and Public Health, Department of Bioanalysis, Faculty of Pharmaceutical Sciences, Ghent University (Ghent, Belgium), which also carried out the determination and quantification of multi-mycotoxins in the extracts. The sample analysis was performed in a Waters^®^ Acquity UPLC system coupled to a Quattro XEVO TQS mass spectrometer obtained from Waters^®^ (Manchester, UK). The software used for data acquisition and processing was MassLynx™ version 4.1 and QuanLynx^®^ version 4.1 also obtained from Waters^®^ (Manchester, UK). The separation of analytes was carried out by a HSS T-3 column (2.1 × 100 mm, 1.8 μm) (Waters^®^, Manchester, UK). Two different mobile phases were used, mobile phase A (water/methanol/acetic acid; 94/5/1, *v/v/v*) and mobile phase B (methanol/water/acetic acid; 97/2/1, *v/v/v*), both buffered with 5 mM ammonium acetate, and were set at a flow rate of 0.3 mL/min. The total run time was 28 min. Five µL of sample was injected into the UPLC system. The liquid chromatography-tandem mass spectrometry multi-analysis method allowed the quantification of 23 toxins including DON, 3-Ac-DON, 15-Ac-DON and 3GDON (Appendix A), as previously reported in the reference method [29]. The method was previously validated showing suitable analytical performance with percentages of recoveries ranging between 100 and 113% for DON, 15-Ac-DON, 3-Ac-DON and 3GDON (Appendix A).

### 4.3. Cereal-Based Food Consumption Data

#### 4.3.1. National-Based Consumption Data

Summary estimates of the general Spanish population consumption data were obtained from the National Dietary Survey on the Child and Adolescent Population in Spain (ENALIA) and the Spanish National Dietary Survey in adults, elderly and pregnant women (ENALIA2) conducted by the Spanish Agency for Consumer Affairs, Food Safety and Nutrition (AESAN) [30]. The design, protocol and methodology of the ENALIA studies following the European Food Safety Agency (EFSA) “EU Menu” guidance recommendations have been described elsewhere [31]. Briefly, ENALIA was conducted between 2012 and 2014 following population-based and representative sampling of the under 18 years old Spanish population. Dietary assessment was based on two non-consecutive 1-day food diaries for children aged 6 months to 10 years, and two 24 h dietary recalls for 11 to 17 year old children and adolescents, separated by at least 14 days to ensure that the information best resembled the usual dietary intake. In turn, ENALIA2 consisted of a dietary survey conducted between 2013 and 2015 in a sample of 900 adults/elderly and 133 pregnant women. The food intake was evaluated with two non-consecutive days (at least 14 days in between), 24 h recalls and a Food Propensity Questionnaire [30].

#### 4.3.2. Regional-Based Consumption Data

In order to obtain individual consumption profiles in the Catalonian region, we conducted a self-administered online Food Frequency Questionnaire (FFQ) for the most consumed cereal-based foods in the region and which may represent a source of *Fusarium* mycotoxins exposure. The semi-quantitative FFQ described elsewhere [11], was completed between 08 May 2019 and 01 of July 2019 by 1564 adult Catalan people divided into four age ranges: 18–25 years old (*n* = 562), 26–40 years old (*n* = 307), 41–60 years old (*n* = 401) and elder people above 60 years old (*n* = 295). The daily consumption data were normalised by the individual weight (kg) and expressed as g of food/kg bw/day. We used the previously published cereal-based infant food consumption data to provide specific exposure estimates in the region [11].

### 4.4. Exposure Assessment

A deterministic method and two probabilistic methods were used in the present work to integrate contamination data with consumption data. For the deterministic method, we combined summary estimates of contamination data and individual consumption data using Equation (1), assuming independency between consumption (*Cπ,j*) and contamination (*T*j) [11].
(1)$${\hat{E}}_{\pi_{0}}=\overset{p}{\underset{j=1}{\sum }}{\bar{C}}_{\pi_{0},\;j}\cdot \bar{T_{j}}$$
where ${\hat{E}}_{\pi_{0}}$ is the normalised exposure of sample population π_0_; ${\bar{C}}_{\pi_{0},\;j}\;$is the arithmetic mean of normalised consumption of food *j* by population π0; and $\bar{T_{j}}$ is the arithmetic mean of contamination of food *j*.

The probabilistic method used in this work consisted of a semi-parametric method (SP) previously described elsewhere [11,32].

In order to conduct the SP approach, we first fitted the probability density function (*pdf*) of consumption data vectors supported by graphical methods and Anderson–Darling statistics. Next, we drew the individual exposure profiles randomly sampling the normalised consumption of food *j* from the respective *pdf*, combined it with the respective level of mycotoxin average contamination of food *j* and integrated it with the individual global exposure equation. We considered the fixed arithmetic mean contamination in the probabilistic models in order to estimate the individual common exposures and related chronic exposure instead of drawing acute exposure profiles. In both probabilistic methods, 10,000 iterations were considered. Both NP and SP methods have been carried out with the EnviroPRA v1.0 package R software [33].

Regarding contamination data, for the adult population we have considered only the upper-bound (UB) scenario to manage left-censored data due to large frequency rates; hence, censored data were substituted by the limit of detection (LOD), as a worst-case scenario. Due to the large number of baby food samples with non-detectable levels (Table 1), we considered three scenarios to address the uncertainty. In addition to the UB scenario, we considered the lower-bound (LB) scenario, where censored data were substituted by zero representing the best-case scenario. A third intermediate scenario was estimated by using the analytical results from the composite of all infant food samples.

### 4.5. Risk Characterisation

To characterise the health risk associated with DON, we divided the exposure daily intake (EDIs) estimates by the reference level of concern, considering the total daily intake (*TDI*) as the threshold as shown in Equation (2):(2)$$\%TDI=\frac{EDI}{TDI}\times 100$$

For chronic dietary exposure to DON, we considered the *TDI* of 1000 ng/kg bw/day based on a clear dose-response for reduced body weight gain in male and female mice with a BMDL05 of 0.11 mg/kg bw/day and an uncertainly factor of 100 [3]. The health-based guidance values (HBGV) of 300 ng/kg bw/day was additionally used for children based on the additional uncertainty factor of 3 considered by the panel of experts on food contaminants from The French Agency for Food, Envi-ronmental and Occupational Health & Safety (ANSES) [9]. For acute exposure, the the acute reference dose (ARfD) of 8000 ng/kg bw/day was used [3].

## Figures and Tables

**Table 1 toxins-14-00506-t001:** Occurrence of deoxynivalenol (µg/kg) in analysed cereal-based foodstuff and their maximum level values allowed by the European Commission Regulation (EC) 1881/2006.

	*n*	DF	Mean	Median	Min	Max	Maximum Level Allowed
White bread	4	100	76.71	79.85	52.81	94.34	500 (Category 2.4.6, Bread (including small bakery wares), pastries, biscuits, cereal snacks and breakfast cereals)
Wholemeal bread	4	100	183.5	177.28	165.6	214.1	
Sliced bread	3	100	166.4	115.4	69.23	314.5	
Bread rusks	3	100	74.95	99.20	10.80	114.9	
Crackers	3	100	28.75	31.44	15.60	39.21	
Cookies	3	66	17.90	21.99	2.50	29.21	
Other cookies	3	66	31.82	25.21	2.50	67.77	
Muffins	3	100	16.91	16.24	14.11	20.39	
Cakes	3	100	30.98	28.17	21.62	43.13	
Breakfast cereals	3	100	66.74	81.62	19.74	98.87	750 (Category 2.4.4, Cereals intended for direct human consumption)
Pasta	3	100	65.53	66.01	63.44	67.15	750 (Category 2.4.5, Pasta (dry))
Baby food	32 *	81	9.10	2.50	2.50	180.8	200 (Category 2.4.7, Processed cereal-based foods and baby foods for infants and young children)

DF, detection frequency (% of samples above the limit of detection). * Individual contamination data of baby food analysed by brands. n, number of analysed composites.

**Table 2 toxins-14-00506-t002:** Normalised consumption (g/kg bw/day) of cereal-based products for the adult population from the Catalonia region and for all of Spain. Mean levels and standard deviation (SD) were computed considering only consumers.

		Catalonia Region				Spain			
Food	Age	N	%	Mean	SD	N	%	Mean	SD
	Group	Total	Cons			Total	Cons		
White bread	18–40	864	61%	1.04	0.64	281	87%	1.17	0.84
	41–60	397	65%	1.23	0.59	209	89%	1.13	0.67
	>60	294	66%	1.26	0.58	313	87%	1.07	0.70
Whole bread	18–40	864	29%	1.03	0.63	281	5%	0.70	0.55
	41–60	397	30%	1.10	0.65	209	12%	1.00	0.81
	>60	294	29%	1.09	0.65	313	12%	0.71	0.66
Sliced bread	18–40	864	32%	0.39	0.27	281	34%	0.68	0.32
	41–60	397	22%	0.36	0.24	209	17%	0.66	0.33
	>60	294	22%	0.30	0.19	313	18%	0.61	0.22
Bread rusks	18–40	864	21%	0.06	0.04	281	2%	0.24	0.21
	41–60	397	26%	0.06	0.04	209	3%	0.21	0.06
	>60	294	25%	0.05	0.04	313	2%	0.24	0.06
Crackers	18–40	864	22%	0.01	0.01	281	5%	0.24	0.16
	41–60	397	18%	0.01	0.01	209	5%	0.21	0.16
	>60	294	15%	0.01	0.01	313	3%	0.21	0.11
Cookies	18–40	864	20%	0.18	0.11	281	24%	0.50	0.34
	41–60	397	26%	0.16	0.09	209	20%	0.46	0.29
	>60	294	37%	0.14	0.10	313	19%	0.36	0.18
Other cookies	18–40	864	38%	0.52	0.37	281	15%	0.75	0.68
	41–60	397	35%	0.45	0.28	209	16%	0.49	0.31
	>60	294	30%	0.39	0.26	313	12%	0.46	0.31
Muffins	18–40	864	19%	0.46	0.32	281	10%	0.68	0.36
	41–60	397	22%	0.52	0.37	209	13%	0.64	0.40
	>60	294	28%	0.41	0.25	313	8%	0.60	0.33
Cake	18–40	864	32%	0.47	0.23	281	7%	0.89	0.44
	41–60	397	27%	0.51	0.35	209	7%	1.19	0.66
	>60	294	26%	0.47	0.27	313	8%	1.01	0.52
Breakfast cereals	18–40	864	34%	0.59	0.15	281	4%	0.48	0.26
	41–60	397	24%	0.53	0.13	209	2%	0.50	0.29
	>60	294	24%	0.50	0.13	313	4%	0.49	0.21
Pasta	18–40	864	85%	0.54	0.20	281	44%	0.81	0.48
	41–60	397	75%	0.45	0.18	209	42%	0.64	0.41
	>60	294	84%	0.40	0.16	313	37%	0.64	0.41

**Table 3 toxins-14-00506-t003:** Exposure to deoxynivalenol (ng/kg bw/day) among the adult population. The estimates were calculated with the deterministic and semi-parametric probabilistic method (SP) through consumption of cereal-based food from the Catalonia region and Spain considering only consumers using the upper-bound scenario for left-censored data.

Method	Region	Age Group	Median	%TDI	Mean	SD	Minimum	P75	P95	P99
Deterministic	Catalonia	18–40	156.3	15.6	182.9	111.7	10.86	235.2	400.7	535.6
		41–60	150.6	15.1	179.1	105.6	1.78	223.0	400.0	515.9
		>60	156.7	15.7	174.8	98.74	14.88	213.1	395.1	469.7
	Spain	18–40	129.4	12.9	157.4	106.7	5.13	218.1	356.8	453.2
		41–60	121.8	12.2	144.7	102.9	1.95	180.3	309.0	542.2
		>60	117.4	11.7	127.6	81.07	1.56	166.8	278.1	393.0
Probabilistic	Catalonia	18–40	152.0	15.2	185.2	132.6	0.00	242.8	434.5	635.1
		41–60	144.4	14.4	179.7	145.5	0.00	237.5	447.5	696.3
		>60	141.2	14.1	177.3	142.1	0.00	231.0	455.0	679.7
	Spain	18–40	134.5	13.4	161.6	127.5	0.00	218.1	391.2	612.8
		41–60	116.2	11.6	167.6	179.5	0.00	204.1	491.5	932.1
		>60	105.5	10.5	143.2	141.9	0.00	185.6	399.8	692.9

Abbreviations: TDI, tolerable daily intake; SD, standard deviation.

**Table 4 toxins-14-00506-t004:** Summary of exposure assessment to deoxynivalenol among adult population reported in previous studies.

Country	Year	Contamination	Group	Consumption Data	Method	Mean Estimated Exposure	Reference
		Data				(ng/kg bw/day)	
Spain	2010	*n* = 526	>65 year	*n* = 76	Deterministic	40 (4.0% TDI)	[11]
					Lognormal	370 (37.0% TDI)	
					Gamma	280 (28.0% TDI)	
			Men	*n* = 336	Deterministic	90 (9.0% TDI)	
					Lognormal	370 (37.0% TDI)	
					Gamma	370 (37.0% TDI)	
			Women	*n* = 384	Deterministic	100 (10.0% TDI)	
					Lognormal	560 (56.0% TDI)	
					Gamma	560 (56.0% TDI)	
France	2008	*n* = 577	Adults	*n* = 1918	Deterministic	373 (37.3% TDI)	[15]
Netherlands	2013	*n* = 1617	7–69 years	*n* = 3819	Probabilistic	95.8–107 (9.6%–10.7% of TDI)	[10,21]
Norway	2010–2011	*n* = 162–185	18–70 years	*n* = 1787	Deterministic	270–470 (27%–47%)	[24]
Czech Republic	2004	*n* = 336	Men	*n* = 711	Deterministic	117–163 *	[23]
						(11.7–16.3% TDI)	
			Women	*n* = 746	Deterministic	81–131 *	
						(8.1–13.1% TDI)	
Tunisia	2017	*n* = 115	Adults		Deterministic	26.2 (2.6% TDI)	[25]
Algeria		*n* = 30 (wheat)	Adults		Deterministic	4920 (492.0% TDI)	[26]
China	2002	*n* = 7356	Adults	*n* = 51,175	Probabilistic	300 (30.0% TDI)	[22]

* Values of P95 obtained by subtraction of beer values.

**Table 5 toxins-14-00506-t005:** Exposure to deoxynivalenol (ng/kg bw/day) among the population under 18 years. The estimates were calculated with the deterministic and probabilistic method through the consumption of cereal-based food from the Catalonia region and Spain considering only consumers.

Method	Region	Age Group	N	Median	%TDI	%HBGV	Mean	SD	Min	P75	P95	P99
Deterministic LB	Catalonia	0–36 months	133	104.8	10.5	34.9	109.0	81.96	2.93	154.2	271.1	349.9
Deterministic UB		0–36 months	133	136.4	13.6	45.5	141.9	106.7	3.82	200.8	353.0	455.5
Deterministic C		0–36 months	133	72.15	7.2	24.1	75.02	56.42	2.02	106.2	186.7	240.9
Deterministic	Spain	0–11 months	182	37.09	3.7	12.4	73.3	87.85	4.97	91.41	240.1	385.8
		12–35 months	333	183.8	18.4	61.3	228.7	199.1	4.71	320.1	537.6	838.8
		3–9 years	589	193.7	19.4	64.6	226.8	142.0	9.1	287.8	514.5	707.8
		10–17 years	627	135.3	13.5	45.1	159.2	100.7	10.11	200.5	362.6	509.1
Probabilistic	Spain	0–11 months	10,000	36.08	3.6	12.0	68.05	80.41	0	97.62	224.6	381.1
		12–35 months	10,000	177.2	17.7	59.1	208.7	150.6	0	280.4	506.0	698.9
		3–9 years	10,000	192.2	19.2	64.1	240.3	192.2	0	287.8	604.3	1007.0
		10–17 years	10,000	133.3	13.3	44.4	176.9	149.7	0	211.9	476.8	780.4

Abbreviations: %HBGV, % health-based guidance value of 300 ng/kg bw/day proposed by ANSES [3]; %TDI, tolerable daily intake; SD, standard deviation; LB, lower-bound; UB, upper-bound; C, composite.

**Table 6 toxins-14-00506-t006:** Summary of exposure assessment to deoxynivalenol among the population under 18 years reported in previous studies.

Country	Year	Contamination Data	Group	Consumption Data	Method	Estimated Mean Exposure	Reference
						(ng/kg bw/day)	
Spain	2016–2017	*n* = 60	4 months		Deterministic	250 (25% TDI)	[19]
			5 months			280 (28% TDI)	
			6 months			320 (32% TDI)	
			7–12 months			360 (36% TDI)	
Catalonia (Spain)	2009	*n* = 90	0–3 years	*n* = 164	Deterministic	740 (74% TDI)	[11]
					Probabilistic	900 (90% TDI)	
France	2005	*n* = 5484	1–4 months	*n* = 124	Deterministic	2.87–59.5	[9]
						(0.29–6% TDI)	
			5–6 months	*n* = 127		101–455	
						(10.1–45.5% TDI)	
			7–12 months	*n* = 195		216–553	
						(21.6–55.3% TDI)	
Norway	2000–2001 2006–2007		1, 2, 4, 13 years old	*n* = 5515	Deterministic	380–2000	[24]
						(38%–200% TDI)	
Portugal	2014	*n* = 52	1–3 years	*n* = 75	Deterministic	108.9–112.8 (10.9%–11.3%)	[18]
					Monte Carlo	53.93 (5.4%)	
	2014–2017	*n =* 963	0–9 years	*n =* 1515	Monte Carlo	(146.2% TDI)	[27]
			10–17 years	*n =* 699		(87.3% TDI)	
Netherlands	2008	*n* = 1617	2–6 years	*n* = 1279	Monte Carlo	280–297 (28%−29.7%)	[21]
Turkey	2018	*n* = 75	<1 year		Deterministic	161 (16.1% TDI)	[20]
Tunisia	2016	*n* = 117	8–12 months		Deterministic	47–279 *	[28]
						(0.47–27.9% TDI)	
						163–979 **	
						(16.3–97.9% TDI)	

* Low consumer data. ** High consumer data, non-available data.

**Table 7 toxins-14-00506-t007:** Inventory of individual samples and composites prepared for the analysis for each food group.

	Individual Samples	Analytical/Composite Samples
White bread	68	4
Whole bread	59	4
Sliced bread	22	3
Bread rusks	20	3
Crackers	21	3
Cookies	21	3
Other cookies *	22	3
Muffins	20	3
Cake	22	3
Breakfast cereals	20	3
Pasta	20	3
Baby food	32	32
Total	347	63

* Includes chocolate cookies, cream cookies, butter cookies, tea cookies and digestive cookies.

## Data Availability

The raw contamination data is available upon request. Spanish food consumption data is property and managed by the Spanish Agency for Consumer Affairs Food Safety and Nutrition, Spanish Minister of Consumer Affairs and Social Welfare.

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
