# Peer review of "Deterministic and Probabilistic Dietary Exposure Assessment to Deoxynivalenol in Spain and the Catalonia Region"

_toxins, 2022, doi:10.3390/toxins14070506_

Round 1
Reviewer 1 Report
This MS addressing one of the most frequently detected mycotoxins worldwide, DON. However, it’s not the most important as Aflatoxins that classified as a group 1 carcinogen to human by IARC. Title: DON is well-known Fusarium mycotoxins and therefore you don’t need to define in the title. The MS is focusing mainly on the dietary exposure of the toxin rather than the occurrence and detection in different food matrix, so please modify the title accordingly. Abstract is lacking of attractively, you should include the most significant finding avoiding use of overstatements and un-necessary details. Likewise for the introduction. Table 1 “Established legal maximum
Contamination values”, is not the right term, you mean the maximum value? Breaking down the type of food must associated with references for those not specified in EC. Using of some terms like “min, max, median…..etc must be presented in a clear way. Tables like 2 and S1 are better to have them as a colored graph rather than text. Please reconsider the conclusion to support your result and overall objectives. The language is appropriate but I feel it must re-checked.
Author Response
Reviewer 1
Comments and Suggestions for Authors
This MS addressing one of the most frequently detected mycotoxins worldwide, DON. However, it’s not the most important as Aflatoxins that classified as a group 1 carcinogen to human by IARC.
Authors’ response: Thank you for your time and your valuable feedback on our draft.
Title: DON is well-known Fusarium mycotoxins and therefore you don’t need to define in the title. The MS is focusing mainly on the dietary exposure of the toxin rather than the occurrence and detection in different food matrix, so please modify the title accordingly.
Authors’ response: Thank you for the suggestion, we have rewritten accordingly, now reads as follows:
“Deterministic and probabilistic dietary exposure assessment to deoxynivalenol in Spain and the Catalonia region”
“Abstract is lacking of attractively, you should include the most significant finding avoiding use of overstatements and un-necessary details. Likewise for the introduction.
Author’s response: we have thoroughly edited the abstract and introduction to improve them according your suggestions
Table 1 “Established legal maximum Contamination values”, is not the right term, you mean the maximum value? Breaking down the type of food must associated with references for those not specified in EC. Using of some terms like “min, max, median…..etc must be presented in a clear way.
Authors’ response: good point, the Table have been updated with your suggestions.
Tables like 2 and S1 are better to have them as a colored graph rather than text.
Author’s response: thank you for the comment but we believe that graphs may become confusing due to the massive information in there, in addition quantitative data in the tables will be helpful for readers intending to use the data for meta-analysis of other types of analysis. In any case, we would be also happy to develop a graph if the reviewer judge that is necessary.
Please reconsider the conclusion to support your result and overall objectives. The language is appropriate but I feel it must re-checked.
Authors’ response: The conclusion has been edited following your suggestion and the language has been deeply checked across all the document.
Reviewer 2 Report
This study concerns the mycotoxin deoxynivalenol (DON) produced by Fusarium fungi. The authors investigated DON and its main metabolites in cereal-based foods and conducted a dietary exposure assessment.
First the mycotoxin, its derivates, occurrence and regulatory efforts towards it are introduced. Relevant literature towards the detection of DON is cited and the uncertainty of the regarded safe levels is presented. Concerning exposure levels found in Spain and France are mentioned. The dietary modeling approach is explained in distinction to other models.
The authors only detected DON itself and no derivatives in levels higher than LOD in their cereal-based samples. They compared their findings in nearly 350 samples with previous studies and found some high outliers, mostly comparable concentrations for bread and some lower concentrations for transformed products and baby food. All samples were compliant with current maximum levels established by European legislation.
This data was than combined with normalized consumption levels of cereal-based products for different age groups to assess the exposure of the population in Spain and Catalonia. By far the main contributor to the exposure was white bread, although exposure levels were not concerning and in line with previous studies and studies from other countries. Only one concerning sample of baby food was found, while exposure levels for infants overall decreased in comparison to previous studies.
The analytical method used was based on samples spiked with DOM as an internal standard and an analysis with LC/MS-MS after extraction.
All in all, this study offers nothing new in terms of methods or results. The methods used were already established and data processing as well as exposure assessment was also known. However, even though the results were not concerning and comparable to past studies, it is still important to regularly monitor contamination and exposure levels in food. Since the last reported study of such for the region of Spain/Catalonia was published already a while ago this work adds value to the literature. It may be used in the future to comprehend historic levels of contamination and exposure. It may also serve as a base for further food safety reflections, intervention strategies and safety thresholds.
Questions for the authors towards the limitations of this study:
1) Why was only one of several regulated and emerging mycotoxins investigated, since multi-mycotoxin methods (also by LC/MS-MS) are established by know?
2) Why were only cereal based products (primary and secondary contamination) and baby food (carry-over contamination) chosen for the investigation? Can other foods not contribute to the exposure level of adults via carry-over contamination and especially nuts/ nut-based products through primary and secondary contamination?
Author Response
Authors’ response: Thank you for your time and your valuable feedback on our draft.
Questions for the authors towards the limitations of this study:
1) Why was only one of several regulated and emerging mycotoxins investigated, since multi-mycotoxin methods (also by LC/MS-MS) are established by know?
Authors’ response: Thanks for pointing this out. Out of 23 mycotoxins analyzed, only DON and Enniatin B were detected in a sufficient number of samples to conduct a proper exposure assessment study. Due to the length and specific interest for the readers, we have conducted the evaluations separately. We have added a clarification in the text.
2) Why were only cereal based products (primary and secondary contamination) and baby food (carry-over contamination) chosen for the investigation? Can other foods not contribute to the exposure level of adults via carry-over contamination and especially nuts/ nut-based products through primary and secondary contamination?
Authors’ response: In our previous study conducted in the same region we established a robust priority criteria of main cereal-based foods that could be contaminated by Fusarium toxins and high contributors to the global exposure (Cited in the text as Cano-Sancho 2011 ref [11]). For that reason, those food items that has low occurrence or very low contribution to the global exposure were excluded for operational/cost reasons.
Reviewer 3 Report
This is a well written article conducted to the occurrence studies and exposure risk assessment of DON and its modified forms (3-AcDON, 15-AcDON and DON3G) presented in cereal-products and consumed by population from Spain, Catalonia. Special attention is attributed to exposure assessment by infants as they are less tolerable to DON and there is need for additional data for these subgroups.
Some minor changes recommended to the authors of this manuscript are as follows:
1) In the first page please indicate the type of the manuscript (Article)
2) Abstract: liquid chromatography tandem mass multi-detection method – please add mass spectrometry and maybe better write multi-analysis
3) Lines 35-36 and elsewhere in the text: the same abbreviations should be used for the DON derivatives: In these lines for 3-AcDON, 15-AcDON 3-Ac-DON, and 15-Ac-DON were used. It is recommended to use 3GDON for the glucoronate form as it is mentioned firstly in the text (Line 50).
4) Please check the significant numbers in Table 1 and other tables.
5) Please check the singular and plural forms within the text (language edition recommended) For example, lines 164-166
6) Check the numeration of the chapters.
7) Supplementary materials are included in the text, they should be included after the text or in separate file. It is hard to read the text as the supplementary data are included within the main manuscript text.
8) In the method section, the mass spectrometry detector should be indicated
9) Did authors checked the matrix effect and other parameters of method validation.
10) Conclusion part: some information repeats the text with the abstract. That is not a good style. Recommended to check both sections of the manuscript and provide corrections.
It is recommended to systematize the result section and the discussion part of the exposure analysis.
Did authors check the recent publication on this topic: https://doi.org/10.1016/j.foodcont.2021.108521 (the reviewer is not author or connected to this paper, but recommends to read it especially within the evaluation of exposure studies.
Author Response
Some minor changes recommended to the authors of this manuscript are as follows:
1) In the first page please indicate the type of the manuscript (Article)
2) Abstract: liquid chromatography tandem mass multi-detection method – please add mass spectrometry and maybe better write multi-analysis
3) Lines 35-36 and elsewhere in the text: the same abbreviations should be used for the DON derivatives: In these lines for 3-AcDON, 15-AcDON 3-Ac-DON, and 15-Ac-DON were used. It is recommended to use 3GDON for the glucoronate form as it is mentioned firstly in the text (Line 50).
4) Please check the significant numbers in Table 1 and other tables.
5) Please check the singular and plural forms within the text (language edition recommended) For example, lines 164-166
6) Check the numeration of the chapters.
7) Supplementary materials are included in the text, they should be included after the text or in separate file. It is hard to read the text as the supplementary data are included within the main manuscript text.
8) In the method section, the mass spectrometry detector should be indicated
Authors’ response: Thank you for these comments numerated above, all of them has been addressed as suggested.
9) Did authors checked the matrix effect and other parameters of method validation.
Authors’ response: this is a good issue raised, yes the method was validated for different matrices based on wheat and corn with little matrix effect noticed, similarly to the composition of our products (the validation details are provided in the original paper and cited in the methodology, DOI:10.1021/jf903859z). The consistency of our results with the results published elsewhere support the robustness of the method for the matrices analyzed
10) Conclusion part: some information repeats the text with the abstract. That is not a good style. Recommended to check both sections of the manuscript and provide corrections.
Authors’ response: thanks for the advice, we have thoroughly edited for clarity
It is recommended to systematize the result section and the discussion part of the exposure analysis.
Authors’ response: the entire text has been thoroughly reviewed and improved, special attention has been paid in the results and discussion section to keep a systematic structure
Did authors check the recent publication on this topic: https://doi.org/10.1016/j.foodcont.2021.108521 (the reviewer is not author or connected to this paper, but recommends to read it especially within the evaluation of exposure studies.
Authors’ response: Thanks for pointing this out, yes actually we found very useful that recent paper to put in context our study and discuss our findings. The paper is reference all along the text with the reference number [2]